# Research on boundary control of vehicle-mounted flexible manipulator based on partial differential equations

**Yuzhi Tang**[ORCID]*

Nantong Institute of Technology, Nantong, Jiangsu Province, China

* junz1982fudan_edu@yeah.net

## Abstract

Vehicle-mounted flexible robotic arms (VFRAs) are crucial in enhancing operational capabilities in sectors where human intervention is limited due to accessibility or safety concerns, such as hazardous environments or precision surgery. This paper introduces the latest generation of VFRAs that utilize advanced soft materials and are designed with elongated structures to provide greater flexibility and control. We present a novel mathematical model, derived using Hamilton's principle, which simplifies the analysis of the arm's dynamic behaviors by employing partial differential equations (PDEs). This model allows us to understand how these arms behave over time and space, classifying them as distributed parameter systems. Furthermore, we enhance the practical utility of these robotic arms by implementing a proportional-derivative (PD) based boundary control law to achieve precise control of movement and suppression of vibrations, which are critical for operations requiring high accuracy. Our approach's effectiveness and practical utility are evidenced by numerical simulations, which verify that our advanced control strategy greatly enhances the performance and dependability of VFRAs in actual applications. These advancements not only pave the way for more sophisticated robotic implementations but also have broad implications for the future of automated systems in various industries.

## 1 Introduction

Vehicle-mounted flexible robotic arms (VFRAs) are engineered to seamlessly integrate with vehicular environments, offering unparalleled flexibility and adaptability. These robotic arms, capable of performing a wide range of complex tasks both inside and outside vehicles—such as in cars, rescue vehicles, and unmanned vehicles—are crucial for precision operations and repetitive tasks where human involvement is impractical or dangerous [1–3]. Their design is crucial for precision operations and repetitive tasks, particularly in scenarios where human involvement is impractical or dangerous. Reflecting the broader industrial application of robotic manipulators, as discussed in [4], VFRAs demonstrate how advanced robotics can significantly enhance operational efficiency and safety across various sectors, thereby playing a pivotal role in modern industrial automation. Due to their unique design, these flexible robotic

**Data Availability Statement:** All relevant data are within the manuscript and its Supporting information files.

**Funding:** The author(s) received no specific funding for this work.

**Competing interests:** The authors have declared that no competing interests exist.

arms are generally soft, allowing them to operate flexibly in narrow or confined spaces. This design often incorporates advanced soft materials or flexible alloys to better adapt to irregular shapes and surfaces. However, the flexible design of these VFRAs also leads to vibration issues, which are more common than in rigid robotic arms. These vibrations can affect the precision of the arm's operations, exacerbate mechanical fatigue, and even lead to operational errors [5–8]. Recognizing this challenge, our research focuses on developing innovative vibration control strategies for VFRAs. Through effective vibration suppression measures, we aim to significantly enhance the performance and reliability of the robotic arm, extend its lifespan, and expand its application potential in various complex environments [9, 10].

Most research on flexible arm control has traditionally relied on ordinary differential equations (ODEs) dynamic models, which are simpler in form and facilitate controller design [11–15]. These models are simple in form and facilitate controller design, but they struggle to accurately capture the distributed parameter characteristics of flexible structures and can sometimes lead to system instability and overflow issues. Compared to ODE models, partial differential equation (PDE) models can more accurately describe the dynamic behavior of flexible structures, thus providing more precise control [16, 17]. In [18], Wang et al. address the complexities, apparent instability, and nonlinearity in traditional quadcopter models by introducing a data-driven system identification technique based on neural ODEs. This approach aims to significantly improve the accuracy and efficiency of dynamical system modeling, providing a novel method for enhancing control strategies in quadcopter applications. Additionally, to address the challenges of flexible manipulators, such as vibration and oscillation, this study proposes a solution using a Linear Quadratic Regulator (LQR) based on full state feedback. The approach was validated through simulations in MATLAB and real hardware testing, proving its effectiveness in enhancing the trajectory tracking accuracy and dynamic performance of the flexible manipulators. In [19], Khan et al. address the control issues of flexible joint manipulators by proposing two theorems based on LQR and nonlinear backstepping control methods. These approaches are not only intriguing but also demonstrate high robustness in the control strategy, as evidenced by simulation results. The findings are significant for the deployment of applications involving flexible robots. In [20], Hang et al. investigated the effects of input saturation, inertia uncertainty, and actuator faults on flexible satellites and proposed an adaptive fault-tolerant control strategy based on a disturbance observer and a fault diagnosis observer. This strategy estimates and suppresses vibration disturbances caused by flexible appendages and actuator faults. Simulation results demonstrated that the proposed method effectively enhances the fault-tolerance performance and control precision of the system. Nonetheless, the model relying on ODEs is not without its drawbacks. These include potential overflows or saturation of control inputs, which can exceed the system's physical constraints, as noted in sources such as [21–23]. Additionally, numerical instabilities might occur, magnifying errors and thereby compromising the accuracy of the control system. In [24], Li and Liu explored the issue of consensus tracking control in nonlinear multi-agent systems described by PDEs. Utilizing HP, they characterized the dynamics of each agent through nonlinear fourth-order PDEs and developed an effective mass-consensus tracking control strategy based on this model. In their research, they constructed a Lyapunov function and applied the extended LaSalle's invariance principle to prove the asymptotic stability of the closed-loop control system. The practicality and effectiveness of this control strategy were validated through numerical simulations. Similarly, Cao et al [25] developed a new boundary controller using PDEs for vibration suppression in flexible single-link manipulators, ensuring system stability and validating the effectiveness through simulation.

There are numerous research methods for controlling VFRAs modeled by PDEs, such as boundary control [26–28], modal methods [29, 30], finite element methods [31–33],

distributed control [34–36], and intelligent control methods like fuzzy logic [37, 38] and neural network control [39, 40]. While each method has its suitable applications and limitations—for instance, boundary control is advantageous for directly manipulating the system's boundary conditions to effectively control the state [41], its implementation may require precise measurement and control of boundary conditions, which can be technically complex. On the other hand, distributed control improves the system's resistance to local disturbances through multi-point control [42, 43]. Yet, it necessitates complex communication and coordination mechanisms to ensure effective synchronization between control nodes, potentially demanding substantial computational resources and high-speed communication networks to maintain real-time efficiency and effectiveness. In the study by Zhou [44], an observer-based adaptive boundary iterative learning control method was developed to handle the challenges posed by input backlash and external disturbances in a dual-link rigid-flexible manipulator. Extensive theoretical analysis and numerical simulations have demonstrated the efficacy of this approach in diminishing vibrations and enhancing precision in tracking performance. Similar to Reference [44], He et al. also utilized boundary control methods to manage a dual-link rigid-flexible wing system, enhancing aircraft maneuverability and adaptability through biomimetic principles. The control strategy developed by their team successfully mitigated wing vibrations and ensured precise angular positioning. The system's stability was confirmed using Lyapunov's method, and numerical simulations further validated the effectiveness of the controller [45]. In contrast to boundary controller designs [46–49], Chen et al. crafted a distributed control system by combining multiple control techniques, effectively tackling the challenges of coordinating multiple flexible robotic arms in demanding settings and maintaining system stability. The effectiveness of this approach was also demonstrated through experimental validation [50]. In [51], Raouf pioneered an adaptive distributed control strategy designed specifically for complex flexible-link robotic manipulators. This cutting-edge strategy enhanced dynamic tracking performance and significantly curtailed system vibrations. Through real-world applications, the approach's global stability and practical utility were thoroughly confirmed, showcasing its effectiveness and resilience in managing demanding control environments.

Based on our comprehensive study of model construction and PDE control methods for VFRAs, the novel contributions of our work include the development of an advanced control strategy to mitigate vibrations more effectively than existing methods. This approach not only improves operational safety and accuracy but also sets a new benchmark for future research in robotic arm applications. Here are the main outcomes of our research:

1. The introduction of a PDE-based modeling approach to simulate the dynamics of VFRAs represents a significant advancement over traditional methods that primarily use ODEs. This approach allows for a more precise modeling of spatially distributed parameters.

2. The development of a PD boundary control strategy based on Lyapunov stability theory is novel in its application to VFRAs, offering a systematic method to enhance vibration control which is critical for precise operations.

3. The extensive validation of our control strategies through simulation sets a new benchmark for the robustness and reliability of control systems in handling real-world disturbances in flexible robotic arms.

The structure of this paper is organized as follows: Section 2 discusses the establishment and problem description of the VFRAs model, using Hamilton's principle to construct the dynamics model of VFRAs. Section 3 first outlines the control objectives of this paper, followed by a detailed explanation of the design of a PD-based boundary controller, and the stability of this controller is verified using Lyapunov functions. Section 4 further demonstrates

the effectiveness of the proposed boundary control strategy through MATLAB/SIMULINK simulation experiments. Finally, Section 5 summarizes the research findings and proposes future research directions. This layout ensures a logical flow of ideas and a clear presentation of the theoretical and practical aspects of the paper.

## 2 Model establishment and problem description of VFRAs

### 2.1 Description of problem

**Remark 1**: To enhance clarity, the document employs the following notations: $(*)_x$ denotes the first derivative of any function $(*)$ with respect to $x$, represented as $\frac{\partial(*)}{\partial x}$. Similarly, $(*)_{xx}$, $(*)_{xxx}$, and $(*)_{xxxx}$ represent the second, third, and fourth derivatives of $(*)$ with respect to $x$, expressed as $\frac{\partial^2(*)}{\partial x^2}$, $\frac{\partial^3(*)}{\partial x^3}$, and $\frac{\partial^4(*)}{\partial x^4}$, respectively. For time derivatives, $(*)_t$ indicates the first derivative with respect to $t$, $\frac{\partial(*)}{\partial t}$, and $(*)_{tt}$ signifies the second derivative, $\frac{\partial^2(*)}{\partial t^2}$. These notations are crucial for precise mathematical formulations and calculations throughout the manuscript.

  **Assumption 1**: The robotic arm's deformation is deemed minimal, enabling the simplification of its mathematical model by linearizing the governing deformation equations. This simplification allows us to overlook nonlinear geometric phenomena such as material hardening or slackening resulting from significant deformations. It is also presupposed that the robotic arm's material is homogeneous and isotropic, indicating uniform material properties throughout all directions.

  **Assumption 2**: It is assumed that both actuators and sensors function flawlessly and without delays. This assumption streamlines both the design and analysis of the control system by eliminating the need to adjust for actuator dynamics or sensor noise.

  The research centers on horizontally moving VFRAs, as illustrated in Fig 1. This single-link arm functions entirely within a horizontal plane, receiving control inputs at its endpoint. The model excludes the influence of gravity, designating *XOY* as the stationary inertial coordinate system and *xOy* as the mobile coordinate system that adjusts in accordance with the movements of the VFRAs.

  The boundary conditions for the VFRAs are defined such that the flexural deflection at the origin is zero at any given time, i.e., $y(0, t) = 0$. Additionally, the rate of bending change at the

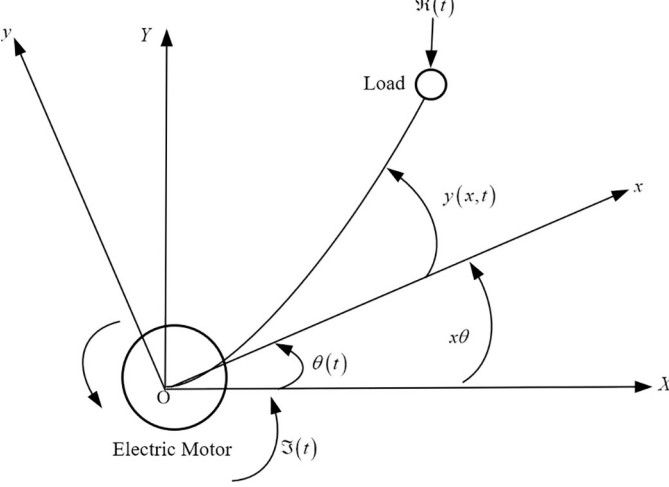

**Fig 1. Schematic diagram of a vehicle-mounted flexible robotic arm.**

origin along the $x$ axis is also zero, which is expressed as $y_x(0, t) = 0$. Therefore, the boundary conditions can be summarized as:

$$y(0) = y_x(0) = 0 \tag{1}$$

Consequently, the position of any point $[x, y(x, t)]$ on the VFRAs in the moving coordinate system $xOy$ can be represented in the inertial coordinate system $XOY$ as:

$$z(x) = x\theta + y(x) \tag{2}$$

where, $z(x)$ denotes the displacement of the VFRAs.

## 2.2 A VFRA model based on PDEs

Based on Eqs (1) and (2), the following conclusions can be drawn:

$$z(0) = 0 \tag{3}$$

$$z_x(0) = \theta \tag{4}$$

$$\frac{\partial^n z(x)}{\partial x^n} = \frac{\partial^n y(x)}{\partial x^n}, (n \geqslant 2) \tag{5}$$

From Eq (5), it follows that

$$z_{xx}(x) = y_{xx}(x), \quad \ddot{z}_x(0) = \ddot{\theta}, \tag{6}$$

$$z_{xx}(0) = y_{xx}(0), \quad z_{xx}(L) = y_{xx}(L), \tag{7}$$

$$z_{xxx}(L) = y_{xxx}(L) \tag{8}$$

**Remark 2**: In this section, HP is utilized to establish the PDE model for the VFRAs. Notably, HP has been extensively applied in the modeling of PDEs, and its effectiveness is well-documented in the literature [52–54]. By accurately calculating the system's kinetic and potential energies along with the virtual work, we efficiently derive the PDEs for the distributed parameter system (DPS). Utilizing this methodology guarantees that the dynamic behavior of the VFRAs is accurately captured, reflecting the complex interplay between its structural flexibility and applied forces, and laying a solid groundwork for subsequent analysis and the development of control strategies.

According to HP [55, 56], the relationship between kinetic energy, potential energy, and the work done by non-conservative forces can be expressed as:

$$\int_{t_1}^{t_2} (\delta E_k - \delta E_p + \delta W_c) \mathrm{d}t = 0 \tag{9}$$

where, the variations in kinetic energy $\delta E_k$, potential energy $\delta E_p$, and the work done by non-conservative forces $\delta W_c$ describe changes in the system's energy. The system's kinetic energy includes the rotational kinetic energy of the joint $\frac{1}{2} I_h \dot{\theta}^2$, the kinetic energy of the entire flexible arm $\frac{1}{2} \int_0^L \rho \dot{z}^2(x) \mathrm{d}x$, and the kinetic energy of the load $\frac{1}{2} m \dot{z}^2(L)$. Therefore, the total kinetic

energy of the system is expressed as:

$$E_{\text{k}} = \frac{1}{2}I_{\text{h}}\dot{\theta}^2 + \frac{1}{2}\int_0^L \rho\dot{z}^2(x)\mathrm{d}x + \frac{1}{2}m\dot{z}^2(L) \tag{10}$$

The potential energy of the VFRAs, caused by bending, is represented as:

$$E_{\text{p}} = \frac{1}{2}\int_0^L \text{EI}y_{xx}^2(x)\mathrm{d}x \tag{11}$$

The work done by non-conservative forces in the system is described by the torque $\tau$ and force $F$ acting at the end of the VFRAs:

$$W_c = \tau\theta + Fz(L) \tag{12}$$

Initially, the expansion of the first term in Eq (9) allowed us to examine the shifts in the kinetic energy of the system. This analysis encompasses the rotational kinetic energy at the joint, the kinetic energy distributed along the length of the VFRAs, and the kinetic energy localized at the load end, which are specifically detailed as follows:

$$\int_{t_1}^{t_2} \delta E_k dt = \int_{t_1}^{t_2}\left(\delta\left(\frac{1}{2}I_{\text{h}}\dot{\theta}^2\right) + \frac{\rho}{2}\int_0^L \delta\dot{z}(x)^2 dx + \delta\left(\frac{1}{2}m\dot{z}(L)^2\right)\right)dt \tag{13}$$

Due to

$$\int_{t_1}^{t_2} \delta\left(\frac{1}{2}I_{\text{h}}\dot{\theta}^2\right)\mathrm{d}t = \int_{t_1}^{t_2} I_{\text{h}}\dot{\theta}\delta\dot{\theta}\ \mathrm{d}t = I_{\text{h}}\dot{\theta}\delta\theta\Big|_{t_1}^{t_2} - \int_{t_1}^{t_2} I_{\text{h}}\ddot{\theta}\delta\theta\ \mathrm{d}t = -\int_{t_1}^{t_2} I_{\text{h}}\ddot{\theta}\delta\theta\ \mathrm{d}t \tag{14}$$

Then

$$\begin{aligned}
\frac{\rho}{2}\int_{t_1}^{t_2}\int_0^L \delta\dot{z}(x)^2\ \mathrm{d}x\ \mathrm{d}t &= \int_0^L \int_{t_1}^{t_2} \rho\dot{z}(x)\delta\dot{z}(x)\mathrm{d}t\ \mathrm{d}x \\
&= \int_0^L\left(\rho\dot{z}(x)\delta z(x)\Big|_{t_1}^{t_2} - \int_{t_1}^{t_2}\rho\ddot{z}(x)\delta z(x)\mathrm{d}t\right)\mathrm{d}x \\
&= -\int_0^L\int_{t_1}^{t_2}\rho\ddot{z}(x)\delta z(x)\mathrm{d}t\ \mathrm{d}x \\
&= -\int_{t_1}^{t_2}\int_0^L\rho\ddot{z}(x)\delta z(x)\mathrm{d}x\ \mathrm{d}t
\end{aligned} \tag{15}$$

where

$$\int_0^L\int_{t_1}^{t_2}\rho\ddot{z}(x)\delta z(x)\mathrm{d}t\ \mathrm{d}x = \int_{t_1}^{t_2}\int_0^L\rho\ddot{z}(x)\delta z(x)\mathrm{d}x\ \mathrm{d}t \tag{16}$$

$$\begin{aligned}
\int_{t_1}^{t_1}\delta\left(\frac{1}{2}m\dot{z}(L)^2\right)\mathrm{d}t &= \int_{t_1}^{t_2}m\dot{z}(L)\delta\dot{z}(L)\mathrm{d}t \\
&= m\dot{z}(L)\delta z(L)\Big|_{t_1}^{t_2} - \int_{t_1}^{t_2}m\ddot{z}(L)\delta z(L)\mathrm{d}t = -\int_{t_1}^{t_2}m\ddot{z}(L)\delta z(L)\mathrm{d}t
\end{aligned} \tag{17}$$

Thus

$$\delta \int_{t_1}^{t_2} E_k \ dt = -\int_{t_1}^{t_2} I_h \ddot{\theta} \delta\theta \ dt - \int_{t_1}^{t_2} \int_0^L \rho \ddot{z}(x)\delta z(x)dx \ dt - \int_{t_1}^{t_2} m\ddot{z}(L)\delta z(L)dt \qquad (18)$$

Next, by unfolding the second term in Eq (9) and applying the condition $z_{xx}(x) = y_{xx}(x)$, we are able to compute the changes in the system's potential energy:

$$
\begin{aligned}
-\delta \int_{t_1}^{t_2} E_p \ dt \ &= -\delta \int_{t_1}^{t_2} \frac{EI}{2} \int_0^L (z_{xxx}(x))^2 \ dx \ dt \\
&= -EI \int_{t_1}^{t_2} \int_0^L z_{xx}(x)\delta z_{xx}(x)dx \ dt \\
&= -EI \int_{t_1}^{t_2} \left( z_{xx}(x)\delta z_x(x)|_0^L - \int_0^L z_{xxxx}(x)\delta z_x(x)dx \right) dt \\
&= -EI \int_{t_1}^{t_2} (z_{xx}(L)\delta z_x(L) - z_{xx}(0)\delta z_x(0))dt \\
&\quad +EI \int_{t_1}^{t_2} \int_0^L z_{xxx}(x)\delta z_x(x)dx \ dt \\
&= -EI \int_{t_1}^{t_2} (z_{xx}(L)\delta z_x(L) - z_{xx}(0)\delta z_x(0))dt \\
&\quad +EI \int_{t_1}^{t_2} \left( z_{xxx}(x)\delta z(x)|_0^L - \int_0^L z_{xxxx}(x)\delta z(x)dx \right) dt \\
&= -EI \int_{t_1}^{t_2} (z_{xx}(L)\delta z_x(L) - z_{xx}(0)\delta z_x(0))dt + EI \int_{t_1}^{t_2} z_{xxx}(L)\delta z(L)dt \\
&\quad -EI \int_{t_1}^{t_2} \int_0^L z_{xxxx}(x)\delta z(x)dx \ dt
\end{aligned}
\qquad (19)
$$

First, expanding the third term of Eq (9) yields:

$$\delta \int_{t_1}^{t_2} W_c \ dt = \delta \int_{t_1}^{t_2} (\tau\theta + Fz(L))dt \qquad (20)$$

Based on the previous analysis, we can express all the relevant variations of kinetic and potential energies along with the work done as follows:

$$
\begin{aligned}
&\int_{t_1}^{t_2} (\delta E_k - \delta E_p + \delta W_c)dt \\
&= -\int_{t_1}^{t_2} I_h \ddot{\theta}\delta\theta \ dt - \int_{t_1}^{t_2} \int_0^L \rho \ddot{z}(x)\delta z(x)dx \ dt - \int_{t_1}^{t_2} m\ddot{z}(L)\delta z(L)dt \\
&\quad -EI \int_{t_1}^{t_2} (z_{xx}(L)\delta z_x(L) - z_{xx}(0)\delta z_x(0))dt + EI \int_{t_1}^{t_2} z_{xxx}(L)\delta z(L)dt \\
&\quad -EI \int_{t_1}^{t_2} \int_0^L z_{xxxx}(x)\delta z(x)dx \ dt + \delta \int_{t_1}^{t_2} \tau\theta + Fz(L)dt
\end{aligned}
\qquad (21)
$$

By applying the boundary conditions $z(0) = 0, z_x(0) = \theta, \ddot{z}_x(0) = \ddot{\theta}$, and the higher-order derivatives $\frac{\partial^n z(x)}{\partial x^n} = \frac{\partial^n y(x)}{\partial x^n} (n \geqslant 2)$, we can further simplify this expression.

$$
\begin{aligned}
&\int_{t_1}^{t_2} (\delta E_k - \delta E_p + \delta W_c) \mathrm{d}t \\
&= -\int_{t_1}^{t_2} \int_0^L (\rho\ddot{z}(x) + \mathrm{EI}z_{xxxx}(x))\delta z(x)\mathrm{d}x \ \mathrm{d}t - \int_{t_1}^{t_2} (I_h\ddot{\theta} - \mathrm{EI}z_{xx}(0) - \tau)\delta z_x(0)\mathrm{d}t \\
&\quad - \int_{t_1}^{t_2} (m\ddot{z}(L) - \mathrm{EI}z_{xxx}(L) - F)\delta z(L)\mathrm{d}t - \int_{t_1}^{t_2} \mathrm{EI}z_{xx}(L)\delta z_x(L)\mathrm{d}t \\
&= -\int_{t_1}^{t_2} \int_0^L A\delta z(x)\mathrm{d}x \ \mathrm{d}t - \int_{t_1}^{t_2} B\delta z_x(0)\mathrm{d}t - \int_{t_1}^{t_2} C\delta z(L)\mathrm{d}t - \int_{t_1}^{t_2} D\delta z_x(L)\mathrm{d}t
\end{aligned}
\tag{22}
$$

where

$$
A = \rho\ddot{z}(x) + \mathrm{EI}z_{xxxx}(x)
\tag{23}
$$

$$
B = I_h\ddot{z}_x(0) - \mathrm{EI}z_{xx}(0) - \mathfrak{I}
\tag{24}
$$

$$
C = m\ddot{z}(L) - EIz_{xxx}(L) - \mathfrak{R}
\tag{25}
$$

$$
D = \mathrm{EI}z_{xx}(L)
\tag{26}
$$

According to HP (9), there are

$$
-\int_{t_1}^{t_2} \int_0^L A\delta z(x)\mathrm{d}x \ \mathrm{d}t - \int_{t_1}^{t_2} B\delta z_x(0)\mathrm{d}t - \int_{t_1}^{t_2} C\delta z(L)\mathrm{d}t - \int_{t_1}^{t_2} D\delta z_x(L)\mathrm{d}t = 0
\tag{27}
$$

Given that the variables $\delta z(x), \delta z_x(0), \delta z(L)$, and $\delta z_x(L)$ are independent within the equation, each term in the expression stands as linearly independent. Consequently, this requires that the coefficients $B$, $C$, and $D$ within the equation be zero. From this analysis, the subsequent PDE dynamic model can be derived as follows:

$$
\rho\ddot{z}(x) = -EIz_{xxxx}(x)
\tag{28}
$$

$$
\mathfrak{I} = I_h\ddot{z}_x(0) - \mathrm{EI}_{xx}(0)
\tag{29}
$$

$$
\mathfrak{R} = m\ddot{z}(L) - \mathrm{EI}z_{xxx}(L)
\tag{30}
$$

$$
z_{xx}(L) = 0
\tag{31}
$$

where

$$
\ddot{z}(x) = x\ddot{\theta} + \ddot{y}(x)
$$

$$
\ddot{z}(L) = L\ddot{\theta} + \ddot{y}(L)
$$

## 3 Design of boundary control law

### 3.1 Control objective

Our control strategy is designed to precisely maneuver Vehicle-mounted Flexible Robotic Arms (VFRAs) to target angular positions $\theta_d$ and velocities $\dot{\theta}_d$, while also effectively suppressing vibrations. We aim to have the position $y(x)$ and velocity $\dot{y}(x)$ along the arm converge to zero. At the end of the arm, we apply a boundary control input $\Re$, guided by a PD control law and a tailored Lyapunov function, to fine-tune vibration damping. This approach emphasizes maintaining stability and accuracy of the robotic arm during operation, thus enhancing the overall system performance and dependability.

### 3.2 Boundary PD control law

In developing the control system, our approach begins by establishing the error and its derivatives related to the angular signal as follows:

$$e = \theta - \theta_{\mathrm{d}}, \tag{32}$$

$$\dot{e} = \dot{\theta} - \dot{\theta}_{\mathrm{d}} = \dot{\theta}, \tag{33}$$

$$\ddot{e} = \ddot{\theta} - \ddot{\theta}_{\mathrm{d}} = \ddot{\theta} \tag{34}$$

We consider the kinetic energy of the VFRAs, the potential energy of the arm, and the kinetic energy of the load. To minimize these energies, particularly the bending deformations $y(x)$, we propose an energy-based Lyapunov function:

$$V = E_1 + E_2 \tag{35}$$

where $E_1$ represents the kinetic and potential energy of the VFRAs, serving as a control index for the suppression of bending deformations and rates of change:

$$E_1 = \frac{1}{2} \int_0^L \rho \dot{z}^2(x)\mathrm{d}x + \frac{1}{2}\mathrm{EI}\int_0^L y_{xx}^2(x)\mathrm{d}x \tag{36}$$

Meanwhile, $E_2$ incorporates the error indices for control and the kinetic energy of the load, reflecting the requirements for control precision:

$$E_2 = \frac{1}{2}I_{\mathrm{h}}\dot{e}^2 + \frac{1}{2}k_{\mathrm{p}}e^2 + \frac{1}{2}m\dot{z}^2(L), \quad k_{\mathrm{p}} > 0 \tag{37}$$

Thus, we monitor the rate of change of the total energy:

$$\dot{V} = \dot{E}_1 + \dot{E}_2 \tag{38}$$

where

$$\dot{E}_1 = \int_0^L \rho \dot{z}(x)\ddot{z}(x)\mathrm{d}x + \mathrm{EI}\int_0^L y_{xx}(x)\dot{y}_{xx}(x)\mathrm{d}x \tag{39}$$

Incorporating the PDE $\rho\ddot{z}(x) = -EIz_{xxxx}(x)$ into our analysis of changes in kinetic and potential energy, we calculate the derivative of $E_1$ as follows:

$$\dot{E}_1 = -EI\int_0^L \dot{z}(x)z_{xxxx}(x)dx + EI\int_0^L y_{xx}(x)\dot{y}_{xx}(x)dx \tag{40}$$

$$\int_0^L \dot{z}(x)z_{xxxx}(x)dx = \int_0^L \dot{z}(x)dz_{xxx}(x)$$

$$= \dot{z}(x)z_{xxx}(x)\big|_0^L - \int_0^L z_{xxx}(x)\dot{z}_x(x)dx = \dot{z}(L)z_{xxx}(L) - \int_0^L z_{xxx}(x)\dot{z}_x(x)dx \tag{41}$$

$$\int_0^L y_{xx}(x)\dot{y}_{xx}(x)dx = \int_0^L z_{xx}(x)\dot{z}_{xx}(x)dx = \int_0^L z_{xx}(x)d\dot{z}_x(x)$$

$$= z_{xx}(x)\dot{z}_x(x)\big|_0^L - \int_0^L \dot{z}_x(x)z_{xxx}(x)dx = -z_{xx}(0)\dot{\theta} - \int_0^L \dot{z}_x(x)z_{xxx}(x)dx \tag{42}$$

By considering the boundary conditions $z_{xx}(L) = 0$ and $\dot{z}_x(0) = \dot{\theta}$, we can further simplify the expression for the rate of change of energy:

$$\dot{E}_1 = -EI\int_0^L \dot{z}(x)z_{xxxx}(x)dx + EI\int_0^L y_{xx}(x)\dot{y}_{xx}(x)dx$$

$$= -EI\left(\dot{z}(L)z_{xxx}(L) - \int_0^L z_{xxx}(x)\dot{z}_x(x)dx\right) + EI\left(-z_{xx}(0)\dot{\theta} - \int_0^L \dot{z}_x(x)z_{xxx}(x)dx\right) \tag{43}$$

$$= -EI\dot{z}(L)y_{xxx}(L) - EIy_{xx}(0)\dot{\theta}$$

Then

$$\dot{E}_1 = -EIy_{xx}(L)\dot{z}(L) - EIy_{xx}(0)\dot{\theta} \tag{44}$$

$$\dot{E}_2 = I_h\ddot{e}\dot{e} + k_p\dot{e} + M_t\dot{z}(L)\ddot{z}(L) = \dot{e}(I_h\ddot{e} + k_pe) + \dot{z}(L)m\ddot{z}(L) \tag{45}$$

Therefore

$$\dot{V} = \dot{E}_1 + \dot{E}_2$$

$$= -EIy_{xxx}(L)\dot{z}(L) - EIy_{xx}(0)\dot{\theta} + \dot{e}(I_h\ddot{e} + k_pe) + \dot{z}(L)m\ddot{z}(L)$$

$$= \dot{e}\left(I_h \cdot \frac{1}{I_h}(\mathfrak{I} + EIy_{xx}(0)) + k_pe - EIy_{xx}(0)\right) \tag{46}$$

$$+\dot{z}(L)(-EIy_{xx}(L) + (EIy_{xxx}(L) + \mathfrak{R}))$$

$$= \dot{e}(\mathfrak{I} + k_pe) + \dot{z}(L,t)\mathfrak{R}$$

The control laws are defined as follows:

$$\mathfrak{I} = -k_pe - k_d\dot{e} \tag{47}$$

$$\mathfrak{R} = -k\dot{z}(L) \tag{48}$$

where $k_p > 0$, $k_d > 0$, $k > 0$, $\Re$ is the boundary control, then

$$\dot{V} = -k_d \dot{e}^2 - k\dot{z}^2(L) \leqslant 0 \tag{49}$$

Given the configurations, we express the system's energy dissipation rate as $\dot{V} = -k_d\dot{e}^2 - k\dot{z}^2(L)$, indicating that the rate of energy loss is non-positive. By employing spatial transformation and using semigroup and compactness analysis methods in this closed-loop system, we facilitate the application of the infinite-dimensional LaSalle Invariant Set Theorem for stability assessment, as supported by recent research [57–59].

### 3.3 Proof and analysis of system stability

Here, we analyze the convergence of the closed-loop system under the condition $\dot{V} \equiv 0$: When $\dot{V} \equiv 0$, it follows that $\dot{e} \equiv 0$ and $\dot{z}(L) \equiv 0$, as well as their second derivatives, $\ddot{e} \equiv 0$ and $\ddot{z}(L) \equiv 0$.

Assuming $\theta_d$ is constant, this implies $\dot{\theta} \equiv 0$ and $\ddot{\theta} \equiv 0$. From the dynamic equation $\rho(x\ddot{\theta} + \ddot{y}(x)) = \rho\ddot{z}(x) = -\mathrm{EI}y_{xxxx}(x)$, we derive:

$$\rho\ddot{y}(x) = -\mathrm{EI}y_{xxxx}(x), \tag{50}$$

$$\rho\ddot{z}(L) = -\mathrm{EI}y_{xxxx}(L) = 0, \tag{51}$$

indicating that $y_{xxxx}(L) = 0$, which leads to $y_{xxx}(L) = 0$.

According to the method of separation of variables, we can assume that the displacement function $y(x)$ can be expressed as the product of two independent variable functions:

$$y(x,t) = X(x) \cdot T(t) \tag{52}$$

where $X(x)$ and $T(t)$ are unknown functions to be determined. From the dynamic equation $\rho\ddot{y}(x,t) = -\mathrm{EI}y_{xxxx}(x,t)$, we can further derive:

$$y_{xxxx}(x,t) = -\frac{\rho}{EI}\ddot{y}(x,t) \tag{53}$$

By substituting the separated variables form, we get $y_{xxxx}(x,t) = X^{(4)}(x) \cdot T(t)$ and $\ddot{y}(x,t) = X(x) \cdot T^{(2)}(t)$. Substituting these into the aforementioned equation, we derive:

$$\frac{X^{(4)}(x)}{X(x)} = -\frac{\rho}{EI}\frac{T^{(2)}(t)}{T(t)} = \mu \tag{54}$$

This implies that the ratio of the two functions is a constant $\mu$, leading to:

$$X^{(4)}(x) - \mu X(x) = 0 \tag{55}$$

Given $\mu = \beta^4$, solving the corresponding differential equation yields the solution for the function $X(x)$ as:

$$X(x) = c_1 \cosh \beta x + c_2 \sinh \beta x + c_3 \cos \beta x + c_4 \sin \beta x \tag{56}$$

where, $c_i$ (where $i$ = 1, 2, 3, 4) are real constants to be determined.

Considering the boundary conditions $y(0) = 0$, $y_x(0) = 0$, $y_{xx}(L) = 0$, and $y_{xxx}(L) = 0$, and utilizing the properties of the differential equation, we find that $X(0)$, $X'(0)$, $X''(L)$, and $X^{(4)}(L)$ should all equal zero. Based on these boundary conditions, the following system of equations is

established to determine the coefficients $c_i$:

$$\begin{cases} c_1 + c_3 = 0 \\ c_2 + c_4 = 0 \\ c_1 \cosh \beta L + c_2 \sinh \beta L - c_3 \cos \beta L - c_4 \sin \beta L = 0 \\ c_1 \cosh \beta L + c_2 \sinh \beta L + c_3 \cos \beta L + c_4 \sin \beta L = 0 \end{cases} \tag{57}$$

By solving the given system of equations, we obtain the following relationships:

$$c_1 \cosh \beta L + c_2 \sinh \beta L = 0, \tag{58}$$

$$c_3 \cos \beta L + c_4 \sin \beta L = 0 \tag{59}$$

$$c_3 \cosh \beta L + c_4 \sinh \beta L = 0, \tag{60}$$

$$c_3 \cos \beta L + c_4 \sin \beta L = 0. \tag{61}$$

Further analysis leads to:

$$c_4 ( \sinh \beta L \cdot \cos \beta L - \sin \beta L \cdot \cosh \beta L) = 0 \tag{62}$$

From this, it is evident that the solution is unique, such that $c_i = 0$ for $i = 1, 2, 3, 4$, thus resulting in $X(x) = 0$, and consequently $y(x) = 0$. Given that $y(x) = 0$ and $y_{xt}(0) = 0$, and considering:

$$I_h \ddot{\theta} = \tau + EIy_{xx}(0) = -k_p e - k_d \dot{e} + EIy_{xt}(0) \tag{63}$$

When $\dot{V} \equiv 0$, it follows that $\ddot{\theta} \equiv 0$, $\dot{e} \equiv 0$, $\ddot{e} \equiv 0$, and $y_{xx}(0) = 0$, thus $e \equiv 0$. Therefore, when $\dot{V} \equiv 0$, we conclude that $e \equiv \dot{e} \equiv y(x) \equiv \dot{y}(x) \equiv 0$, indicating that the closed-loop system is asymptotically stable. As time progresses $t \to \infty$, for all $x \in [0, L]$, $\theta$ will converge to $\theta_d$, $\dot{\theta}$ will converge to $\dot{\theta}_d$, and both $y(x)$ and $\dot{y}(x)$ will approach zero.

## 4 Simulation verification

### 4.1 Simulation parameter setting

In our study, we conducted simulations using MATLAB/Simulink version 2022b, which supports our simulation code. Please note that earlier versions than 2022b might encounter compatibility issues. The simulations ran on a computer equipped with Windows 10 Professional, powered by an Intel(R) Core(TM) i7–14700KF processor at 3.40 GHz, and equipped with 32.0 GB of RAM. This setup was chosen to ensure efficient and stable operation throughout various simulation scenarios, providing the necessary processing power and memory capacity to manage the demanding computational tasks and data processing requirements of our simulations effectively, thereby facilitating uninterrupted and reliable results.

The dynamics of our VFRA model are governed by the equations detailed from (28) through (30). We configured the model parameters as follows: the density $\rho$ is set at 0.2, the product of elastic modulus and moment of inertia EI is 2, the mass $m$ is 6.88kg, the arm length $L$ measures 1.2m, and the moment of inertia $I_h$ is 0.014. We set the target angle to $\theta_d = 0.50$. The control laws implemented are derived from Eqs (47) and (48), with control parameter values of $k_p = 25$, $k_d = 30$, and $k = 10$. For the simulation, we divided space and time into $nx = 10$

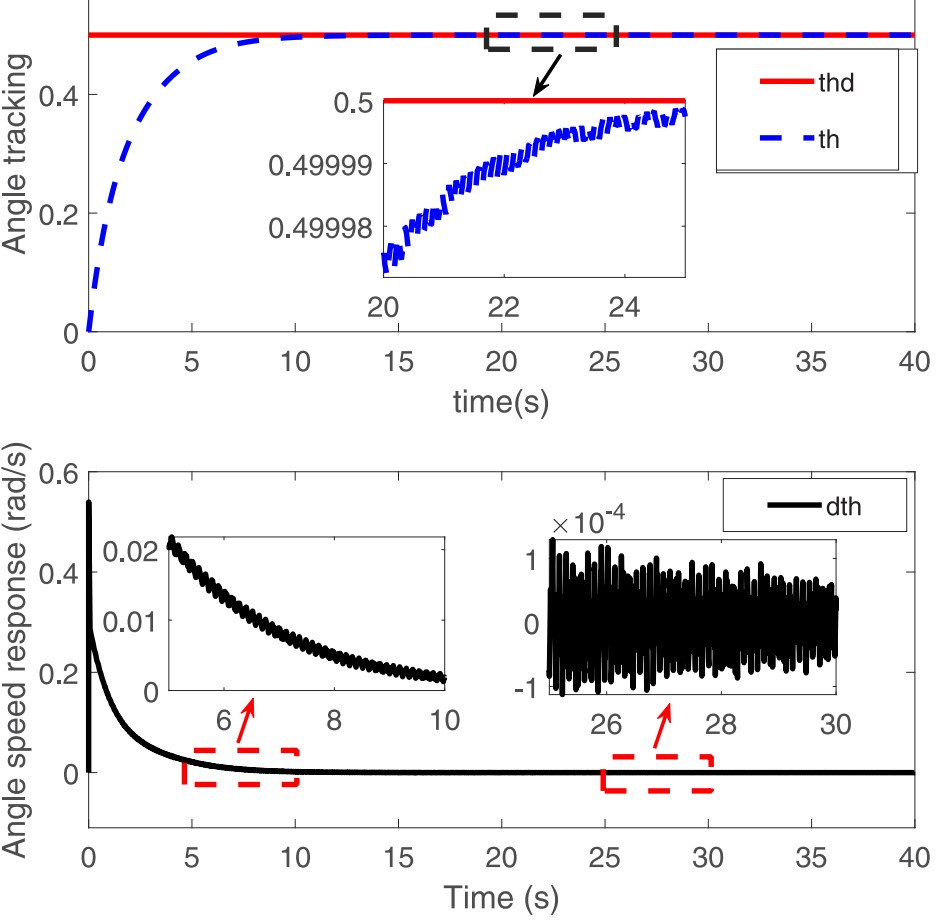

**Fig 2. Angle and angular velocity response.**

and $nt$ = 80000 segments, respectively, spanning a total duration of 40 seconds. The outcomes of these simulations are illustrated in Figs 2 through 7.

## 4.2 Simulation results show

Fig 2 displays the data graphs for angle and angular velocity responses. In the top graph of Fig 2, the red solid line (labeled "thd") represents the target angle, indicating the ideal or desired angular position. The blue dashed line (labeled "-th") shows the actual angle response. Initially, there is a noticeable discrepancy between the actual angle and the target angle as depicted by the blue and red lines not coinciding. However, as the control system adjusts, the actual angle gradually converges to the target, ultimately stabilizing close to the desired setting. This demonstrates the control strategy's effectiveness in correcting deviations over time. The initial response time and the extent of overshoot are areas that could be explored further to enhance performance.

The bottom graph of Fig 2 illustrates how the angular velocity (i.e., the rate of change of angle) varies over time. The angular velocity initially exhibits a rapid decline from a high positive value to near zero, transitions through a negative phase, and finally stabilizes near zero.

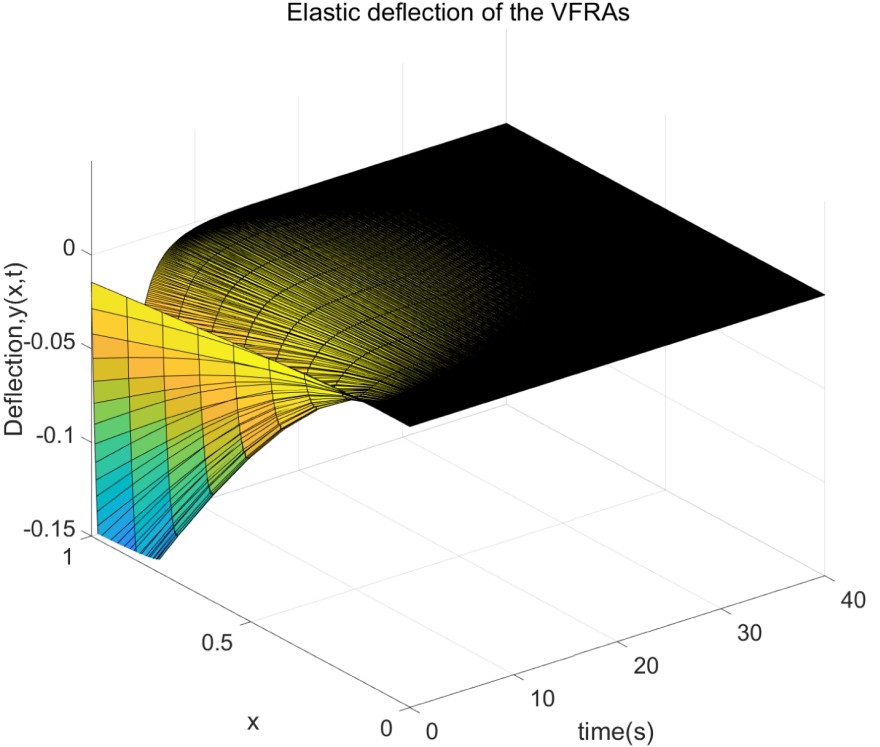

**Fig 3. Three-dimensional graph of distributed elastic deformations on a VFRA.**

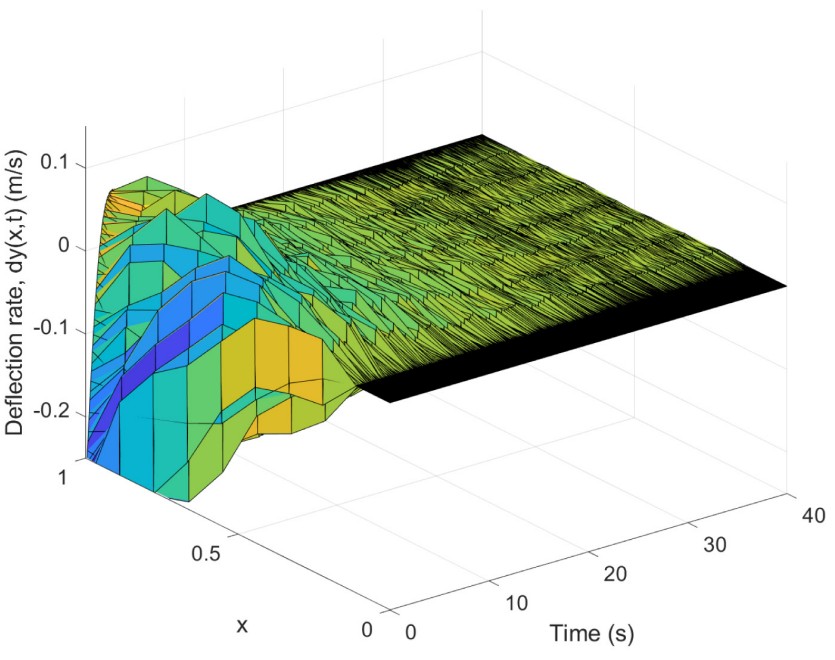

**Fig 4. Three-dimensional graph of the rate of change in distributed elastic deformations on a VFRA.**

This portion of the graph, particularly the zoomed-in view between 26 and 30 seconds, highlights the fluctuations in angular velocity before stabilization, suggesting a critical examination of damping characteristics and response time of the control system to dynamic changes.

Fig 3 showcases the progression of elastic deformation in VFRAs at various time intervals. This plot vividly illustrates the temporal evolution of deformation, where minimal initial deformations intensify notably over time, with pronounced changes at specific positions along the x-axis. Such information is critical as it highlights how external forces dynamically influence the structural integrity of the manipulator. Conversely, Fig 4 presents a three-dimensional visualization of the deformation rate in VFRAs, offering insights into how quickly the material responds to applied forces. The slow initial deformation rate that sharply accelerates at certain positions along the x-axis can inform the design of more resilient flexible manipulators.

Fig 5 presents a comparative analysis of the elastic deformation of VFRAs at two key positions, x = 0.5L and x = L, where L is the total length of the VFRAs. The stark contrast in deformation patterns between these two points is evident; deformation at the midpoint (0.5L) is significantly greater and more intense than at the endpoint (L). This observation could have substantial implications for the design and utilization of VFRAs, indicating potential areas of vulnerability and the need for targeted reinforcement or control adjustments. Figs 6 and 7 provide detailed visualizations of the control system's input signals over time, emphasizing the system's responsiveness to disturbances and its ability to maintain stability under dynamic

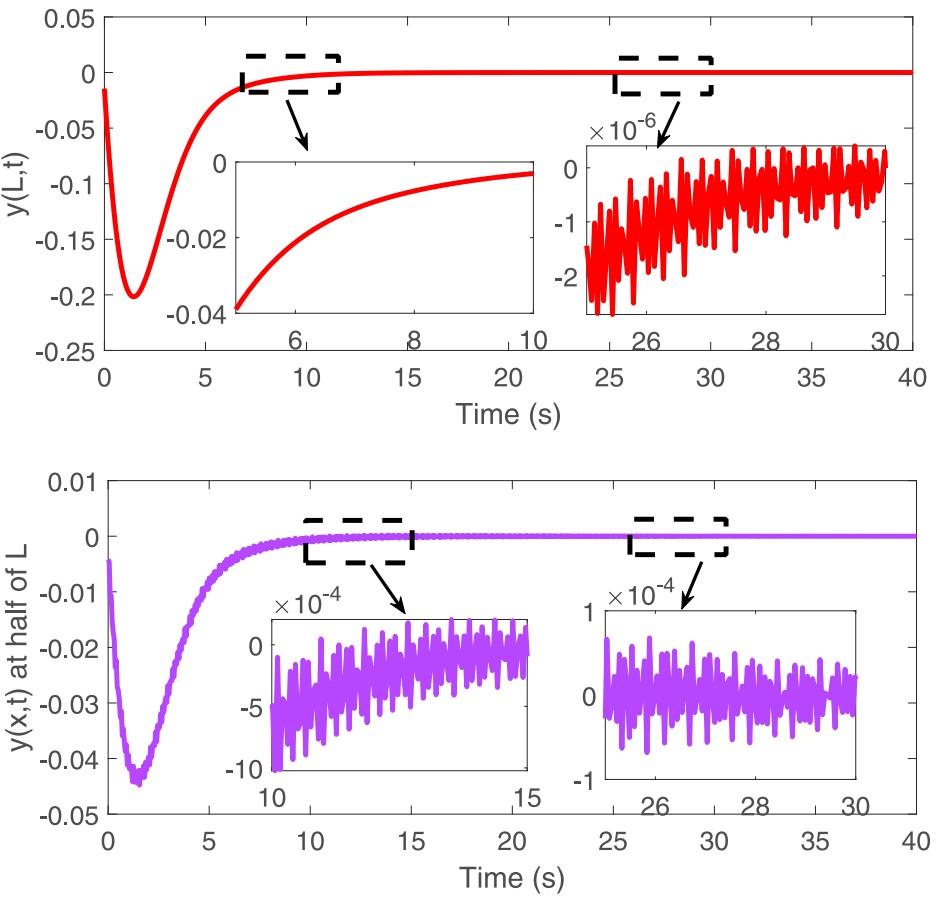

**Fig 5. The schematic diagram of elastic deformation of the VFRA at *x* = 0.5*L* and *x* = *L*.**

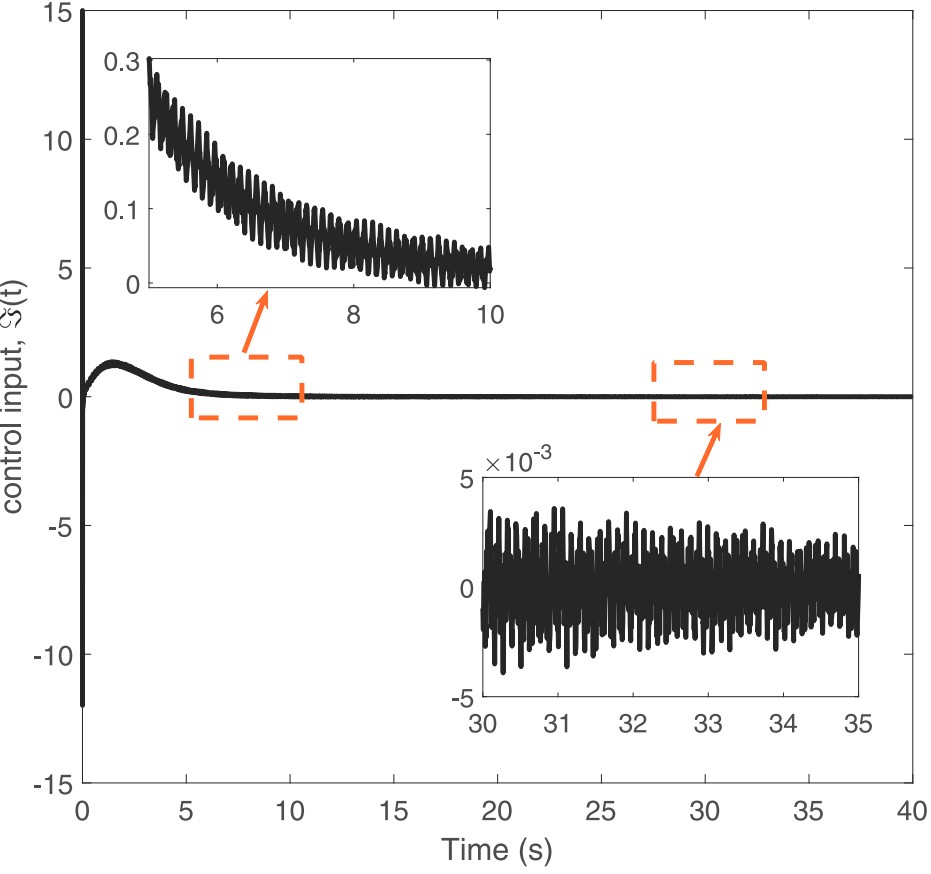

**Fig 6. The control input $\Im(t)$.**

conditions. These graphs are instrumental in understanding the oscillatory behavior and stability of the control system post-adjustment, serving as a foundation for further optimization of the control strategy.

## 5 Discussion and future work

This study developed a dynamic modeling framework and control strategy for VFRAs using PDEs. By applying Hamilton's principle, we established a dynamic model that derives the system's governing PDEs, streamlining the analysis of the complex forces impacting the arm. Additionally, a PD boundary control strategy was designed to achieve precise angle tracking and vibration suppression along the arm. The efficacy of this approach was confirmed through numerical simulations, which highlighted its capability to maintain stability, enhance control accuracy, and diminish vibrations. These contributions provide a robust foundation for advancing the modeling and control of flexible robotic systems, improving their applicability in complex and dynamic environments.

Looking ahead, future research will aim to boost the real-time response of the dynamic model by integrating cutting-edge computational methods, including machine learning and data-driven techniques. To improve the adaptability of the control strategy, we plan to design and test adaptive algorithms capable of handling varying environmental conditions and task requirements. Furthermore, reducing control input delays and mitigating uncertainties will be

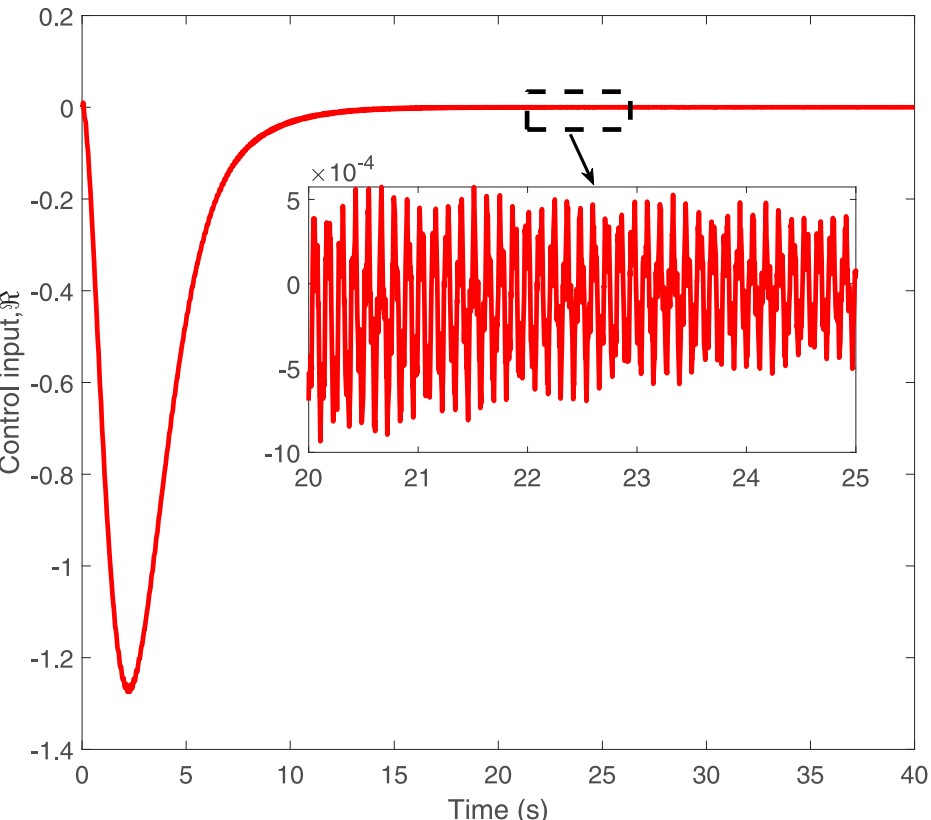

**Fig 7. The control input** $\Re(t)$**.**

addressed by integrating predictive control frameworks and optimization-based algorithms, aiming to enhance system stability and reliability. These initiatives are expected to broaden the range of applications for VFRAs in challenging environments and ensure their robust performance under increasingly stringent conditions.

## Supporting information

**S1 File. Paper program.**
(PDF)

## Author Contributions

**Data curation:** Yuzhi Tang.

**Methodology:** Yuzhi Tang.

**Writing – original draft:** Yuzhi Tang.

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
