## [Decision Letter · Decision Letter 0]

27 Nov 2024

PONE-D-24-52674Research on boundary control of vehicle-mounted flexible manipulator based on partial differential equationsPLOS ONE

Dear Dr. Tang,

Thank you for submitting your manuscript to PLOS ONE. After careful consideration, we feel that it has merit but does not fully meet PLOS ONE’s publication criteria as it currently stands. Therefore, we invite you to submit a revised version of the manuscript that addresses the points raised during the review process.

We look forward to receiving your revised manuscript.

Kind regards,

Zeashan Hameed Khan, Ph.D.

Academic Editor

PLOS ONE

4. We are unable to open your Supporting Information file [S1.zip]. Please kindly revise as necessary and re-upload.

Additional Editor Comments:

The paper needs serious improvements to clearly state the novelty of the research in comparison with the existing literature.

In addition, the paper language also needs improvement.

Reviewers' comments:

Reviewer's Responses to Questions

**Comments to the Author**

1. Is the manuscript technically sound, and do the data support the conclusions?

Reviewer #1: Partly

Reviewer #2: Partly

2. Has the statistical analysis been performed appropriately and rigorously? 

Reviewer #1: I Don't Know

Reviewer #2: N/A

3. Have the authors made all data underlying the findings in their manuscript fully available?

Reviewer #1: Yes

Reviewer #2: No

4. Is the manuscript presented in an intelligible fashion and written in standard English?

Reviewer #1: Yes

Reviewer #2: Yes

5. Review Comments to the Author

Reviewer #1: - The work is limited to a simulation environment. What is the evidence that the proposed control scheme would work exactly in the same way on a real vehicle-mounted flexible robotic manipulator as demonstrated in simulation? Please include the experimental results, be if they are of preliminary nature.

- In the Introduction, briefly elaborate on the pivotal role of robotic manipulators in industry with reference to 'An autonomous image-guided robotic system simulating industrial applications'.

- Please fix the mathematical symbols appearing in the Abstract.

- The crucial role of control in flexible robotic manipulators discussed in Section 1 (Introduction) could benefit from the literature such as 'On the derivation of novel model and sophisticated control of flexible joint manipulator'.

- Please change "...and colleagues" to et al.

- Include MATLAB/Simulink version used to run the Simulations. Also, please mention the specifications of the machine on which simulations were performed.

- There are some interesting results in Section 4, however, the discussion on them needs to be more critical nd conclusive.

- Include results on error parameters such as IAE, ITAE and ISE etc.

Reviewer #2: 1. The abstract delves heavily into technical details, such as mathematical derivations and control laws, without providing sufficient context or emphasizing the broader significance of the study for practical applications or the field as a whole.

2. While the abstract describes the proposed modeling and control strategies, it does not mention how these methods compare to existing approaches, leaving the reader unclear about the novelty or advantages of the work.

3. The introduction should clearly conclude with a distinct section highlighting the novel contributions of your work.

4. The literature review should benefit from more explorations of previous studies. This will provide a richer context and demonstrate how your work builds upon or diverges from established research.

5. The discussion section needs to be expanded to more thoroughly analyze the results.

6. The first paragraph of the conclusion should succinctly summarize the contributions of the study in past tense, clearly stating what has been accomplished and the impact it has on the field.

7. The second paragraph of the conclusion should provide clear and actionable future recommendations.

8. Discussion should be expanded, future work is with conclusion in a new section.

6. PLOS authors have the option to publish the peer review history of their article (what does this mean?). If published, this will include your full peer review and any attached files.

Reviewer #1: No

Reviewer #2: **Yes: **Luttfi A. Al-Haddad

---

## [Author Response · Author response to Decision Letter 0]

6 Dec 2024

The reviewer's reply has been uploaded to the system.

---

## [Decision Letter · Decision Letter 1]

13 Dec 2024

PONE-D-24-52674R1Research on boundary control of vehicle-mounted flexible manipulator based on partial differential equationsPLOS ONE

Dear Dr. Tang,

Thank you for submitting your manuscript to PLOS ONE. After careful consideration, we feel that it has merit but does not fully meet PLOS ONE’s publication criteria as it currently stands. Therefore, we invite you to submit a revised version of the manuscript that addresses the points raised during the review process.

We look forward to receiving your revised manuscript.

Kind regards,

Zeashan Hameed Khan, Ph.D.

Academic Editor

PLOS ONE

Journal Requirements:

Additional Editor Comments:

The paper is in a good shape after the revision and can be accepted. Please use high resolution images in the final version. Also, revise for any language related errors.

Reviewers' comments:

Reviewer's Responses to Questions

**Comments to the Author**

1. If the authors have adequately addressed your comments raised in a previous round of review and you feel that this manuscript is now acceptable for publication, you may indicate that here to bypass the “Comments to the Author” section, enter your conflict of interest statement in the “Confidential to Editor” section, and submit your "Accept" recommendation.

Reviewer #1: All comments have been addressed

Reviewer #2: (No Response)

2. Is the manuscript technically sound, and do the data support the conclusions?

Reviewer #1: Yes

Reviewer #2: (No Response)

3. Has the statistical analysis been performed appropriately and rigorously? 

Reviewer #1: I Don't Know

Reviewer #2: (No Response)

4. Have the authors made all data underlying the findings in their manuscript fully available?

Reviewer #1: Yes

Reviewer #2: (No Response)

5. Is the manuscript presented in an intelligible fashion and written in standard English?

Reviewer #1: Yes

Reviewer #2: (No Response)

6. Review Comments to the Author

Reviewer #1: The authors have addressed all the changes suggested earlier. The paper can be accepted. Please include high resolution images in the final paper.

Reviewer #2: Minor English proofreading is required now that the authors have made the major revision, the comments about the conclusion, please do both.

7. PLOS authors have the option to publish the peer review history of their article (what does this mean?). If published, this will include your full peer review and any attached files.

Reviewer #1: No

Reviewer #2: **Yes: **Luttfi A. Al-Haddad

---

## [Author Response · Author response to Decision Letter 1]

17 Dec 2024

See the uploaded file for the reply of reviewers' opinions.

---

## [Editor Report · Decision Letter 2]

20 Dec 2024

Research on boundary control of vehicle-mounted flexible manipulator based on partial differential equations

PONE-D-24-52674R2

Dear Dr. Tang,

We’re pleased to inform you that your manuscript has been judged scientifically suitable for publication and will be formally accepted for publication once it meets all outstanding technical requirements.

Kind regards,

Zeashan Hameed Khan, Ph.D.

Academic Editor

PLOS ONE

Additional Editor Comments (optional):

Thank you for submitting all the necessary corrections. The paper can be accepted in the present form.
---

## [Editor Report · Acceptance letter]

27 Dec 2024

PONE-D-24-52674R2 

PLOS ONE

Dear Dr. Tang, 

I'm pleased to inform you that your manuscript has been deemed suitable for publication in PLOS ONE. Congratulations! Your manuscript is now being handed over to our production team.

Kind regards, 

on behalf of

Dr. Zeashan Hameed Khan 

Academic Editor

PLOS ONE